# Toward a Socio-Political Approach to Promote the Development of Circular Agriculture: A Critical Review

**DOI:** 10.3390/ijerph192013117

**Published:** 2022-10-12

**Authors:** Chenyujing Yang, Yuanyuan Zhang, Yanjin Xue, Yongji Xue

**Affiliations:** School of Economics and Management, Beijing Forestry University, Beijing 100083, China

**Keywords:** circular agriculture, top-down implementation, environmental sustainability, agroecology, agricultural regime

## Abstract

Under the dual pressure of ensuring global food security and coping with the effects of climate change, many countries have proposed projects of circular agriculture to mitigate the vulnerability of agricultural systems. However, due to the different utilizations of agricultural resources in different countries, there are still some important limitations and obstacles to the promotion of agricultural recycling technologies. This review discusses global circular agriculture projects from a social science perspective. We found that (1) current research on circular agriculture is concentrated in the field of natural sciences with a focus on technological upgrading, neglecting social, political and economic research; (2) top-down circular agriculture projects rely on infrastructure and technical inputs for management, which undermines the focus on public participation and is limited by the timing and intensity of state intervention; (3) the development model led by enterprises or associations relies on cooperation and benefit games with farmers, and its sustainability depends largely on changes in the regulation of the agricultural waste utilization system. Based on this, this review argues that circular agriculture projects are not only technical issues in the field of natural sciences, but also strongly influenced by social development. For future research, we strongly recommend cross-disciplinary cooperation, not limited to technology development.

## 1. Introduction

The COVID-19 pandemic is regarded as an alarm bell by many people, and it has incorporated many concerns regarding social vulnerability in the food system [1,2]. Although the food-oriented goals of the United Nations (UN) Sustainable Development Goals have been in place for six years already since 2014, global food insecurity has been growing. The COVID-19 pandemic propelled world hunger in 2020, which increased from 8.4 to as much as 10.4 percent of the global population in just one year, after remaining virtually stagnant for five years [3]. At the same time, several studies show that current global trends in agricultural carbon emissions will hinder the achievement of the 1.5 °C target, threatening the 2 °C target by the end of the century [4]. Therefore, under the premise of ensuring food security, how to transform agricultural systems from a linear production model toward a more sustainable direction has become a world concern.

Worldwide, the application of the circular economy in agriculture has been a highlight [5]. In order to achieve sustainable development, in the early 1990s, the concept of the circular economy was developed to describe how the economy is affected by natural resources. The goal of a circular economy is to minimize or eliminate the input of nonrenewable materials in the production system and to maximize the reuse of these materials in the same system [6]. The traditional crude linear development model of agriculture (i.e., the “resource–product–waste” model) not only results in a huge waste of resources, but also is a huge source of pollution [7]. For example, in India, when large amounts of paddy straw are burned in agricultural fields, it creates approximately 25% of the straw surplus and contributes to 0.05% of greenhouse gas emissions [8,9]. Circular economy practices can provide opportunities to reduce greenhouse gas emissions from the agricultural sector through the recycling of raw materials, agricultural waste and fertilizers [10]. In addition, bioenergy can be a major partner in the sector through biogas, clean power generation, heat supply, bromoethane production and biofertilizer production [11]. With the increasing challenges in the field of global energy and climate change [12,13,14], the development of bioenergy and reduction in carbon emissions have become common themes in the development of circular agriculture in various countries.

The primary aim of this review was to systematically review the literature to identify the main factors and areas for further research in circular agriculture. Then, we hope to determine the important role of these factors or research directions through typical cases, which may include politics, market, benefits, etc. Finally, we call for a more critical solution from a sociopolitical perspective.

## 2. Circular Agriculture

With the gradual understanding of the economic and environmental benefits of resource recycling, the utilization of waste recycling has attracted people ‘s attention. For decades, researchers and farmers have been discussing the idea of recycling valuable final products into the circular economy by creating a closed loop so as to minimize harmful inputs and outputs in any production system [15]. Some researchers also use similar terms to refer to cycles and apply them to agriculture, such as circular bioeconomy [16,17], rural ecological economy [18], agroecology [19,20] and climate-intelligent regenerative agriculture [21]. These terms aim to transform the traditional linear economy into a circular economy in agriculture. This model usually has the functions of building multifunctional landscapes, enhancing sustainability, supporting agricultural digitization and encouraging changes in eating habits [22]. Circular agriculture may be mainly related to animal manure as fertilizer (cattle, pigs, poultry, etc.). With the development of a circular economy, new methods of generating electricity from agricultural waste have appeared in recent years. In particular, biogas is the core of the cycle model [5].

In fact, many countries in the world have tried and applied a circular economy in agriculture, known as circular agriculture (CA). CA is usually biorefined by agricultural waste sources (such as straw, husk, and stoves of corn and rice residues) to minimize dependence on crude oil [23]. Some locations around the world have been taking measures to utilize biogas, take advantage of environmental advantages and create value from waste [24,25,26]. Many countries, including those in Europe, Africa [27], Asia [28] and North America [29], have taken into account the development of a circular bioeconomy [30]. Among them, China and the EU are the main supporters [31]. Since 2013, China has formulated a national circular economy strategy and made great progress in improving resource utilization efficiency. Since 2015, China has been implementing the national sustainable agriculture plan [32]. In 2018, the EU issued an agricultural policy, from agriculture to agriculture, including a comprehensive set of CA practices [33]. One such example was recently identified by the World Economic Forum (WEF) as one of the Shaping the Future of Food initiatives aimed at developing sustainable, efficient and nutritious food systems in the agricultural sector [34].

Due to the social, political and economic differences, the development models and objectives of circular agriculture vary from country to country. For example, China and Peru have a typical top-down management model, where circular agriculture is government led. In Peru, circular agriculture in border areas is implemented through government legislation to achieve a synergistic transformation between agriculture and bioenergy [35]. In particular, biogas production is integrated into the circular agriculture system to achieve multiple benefits such as agricultural progress, poverty management and carbon emission reduction. The development of circular economy in Italy and Germany is more market dependent. Italy, based on the EU’s circular economy plastics strategy, uses traditional market instruments, namely, subsidies and tax credits as well as initiatives, such as the Extended Producer Responsibility scheme (EPR) [36], to reduce plastic waste and carbon emissions in agriculture.

Many studies have integrated and analyzed many aspects of circular agriculture by review. According to a literature review from 2005 to 2012, the key words used in circular agriculture research include sustainable development, sustainability, recycling, agricultural robots, waste management, and biomass [37]. Some scholars have reviewed the technical knowledge on agricultural residue management [38,39,40] and found that anaerobic digestion and composting is the most explored technology in the scientific literature [41], and poultry manure composting is a potential recycling technology [42,43]. A review of a 2003–2019 study found that factors affecting farmers’ participation in organic recycling agriculture may include environmental attitudes, economic attitudes, age, education and their sources of information [44].

## 3. Research Method and Steps

### 3.1. Research Method

We used Mixed Methods in this review. Words, pictures and dialogues can be used to complement the meaning of numbers under mixed methods, while numbers can be used to improve the accuracy of words, pictures and dialogues. More importantly, the mixed research approach is no longer limited by a single method. It not only answers broader and more complex research questions, but also addresses understandings and insights that are overlooked in a single approach.

Although existing quantitative research methods have addressed many technical aspects of circular agriculture, the issue of the direction is still not well resolved by quantitative research or qualitative research alone. In fact, mixed methods research allows us to gain a more integrated perspective to sort out the development of circular agriculture. We have combed and integrated the existing literature on circular agriculture with quantitative studies. At the same time, based on some secondary data analysis and typical case studies, more open information is obtained, giving more details about the context and experience.

### 3.2. Research Steps

We adopted the following overall research steps:

Step 1: Identify research questions. We defined the following research questions: (1) What are the current research priorities of circular agriculture? (2) What are the gaps between these research priorities and the actual development of circular agriculture in various countries? (3) What new perspective should these gaps be viewed from?

Step 2: Systematic review. In order to understand the current research situation in the field of circular agriculture, we summarized the research from 2000 to 2022 to understand the current situation and to facilitate data collection. We collected all the research in the CA field included in the Web of Science database to analyze its characteristics. This method focused on the qualitative and systematic review of the literature published in English including the determination of all publications in this research field. This type of review is generally used to review information on specific topics and to make readers aware of researchers’ research processes [45]. We reviewed the research on CA including the key words, countries and the highest production institutions. In addition, in order to complete the research, we used the VOSviewer tool to produce network maps;

Step 3: Case selection. To answer our research questions, we used several country cases to analyze the actual development of CA. Due to the large differences in the natural conditions of various countries and regions, there are diverse methods to deal with the problem of agricultural sustainability. We selected CA schemes from China, Germany and Italy with different political backgrounds;

Step 4: Data collection. The data for our study came from the literature review. Firstly, we evaluated the development model of top-down circular agriculture, using China as an example. In order to compare the differences between circular agriculture cases with different political and social backgrounds, we also reviewed the international literature. Finally, we found evidence to prove the importance of the social economy through the literature review.

Step 5: Comprehensive analysis. Based on the above literature review and actual development description, we analyzed these research results through a socio-political approach.

## 4. Current Research on CA Focuses on the Field of Natural Science with Technological Upgrading as the Main Focus

In this review, Boolean operators were used to analyze the papers related to CA in the Web of Science (WoS) core database from the past 20 years, focusing on finding articles and reviews published in journals with high impact factors in order to better ensure the rigor of the content and to grasp the development direction of the research topic. The search date was set from January 2000 to June 2022, with a time span of nearly 22 years. Studies and reviews were included in the literature search.

We excluded a part of the literature with low correlation including repetitive literature and literature with titles unrelated to the research topic. Finally, 8085 articles were selected for analysis. In order to better clarify the similarities and characteristics among the papers, we conducted a deep analysis of the four characteristics of keywords, disciplines, publications and countries. In addition, the VOSviewer software was used to complete the production of various feature relationship maps. The effectiveness of this tool has been satisfactorily verified [46,47].

### 4.1. Biological Resources, Water Resources, Land Resources and Technology Are the Key Words in CA Research

The co-occurrence map of keywords was based on text data settings, and the keywords came from the titles and abstracts. We used full counting to ensure that a term appeared at least 100 times. It can be seen from the figure that scholars mainly focused on circular economy and agriculture when exploring this field. After analysis, we obtained four clusters, which are represented by different colors (Figure 1). Each cluster represents an interesting topic within the scope of this study’s analysis. The relationship between terms reflects the similarity between concepts. Among the groups with green labels, the words related to resources, such as plant, biodiversity, species, forest and evolution, appeared frequently. In the blue marked group, the research object was water resources, which was composed of recovery, cost, efficiency, wastewater and other words emphasizing the reuse of water resources in agricultural production. In the yellow group, the main research object was land, which was composed of fertilizer, organic matter, and health, emphasizing the importance of the sustainable development of land resources. In the red group, the farm is the main body of the implementation of circular agriculture, and circular economy, food security, sustainable development and other words are the goal of circular agriculture technology upgrading.

In general, biological resources, water resources, land resources and technology are the main keywords of CA research. This may be related to CA, as is a sustainable development model proposed under the background of serious shortage of resources and damage to the environment in the process of industrialization. Resource endowment is the basic condition for the development of CA [48]. At the same time, we discovered the importance of water resources management and land resources management in circular agriculture from the keywords, which is similar to the existing research [49].

### 4.2. A large Number of Articles on CA Are Concentrated in the Field of Natural Science

In this review, the literature on the field of circular economy was divided into subject categories, and the top five subjects were selected. It can be seen from Figure 2 that all of the five disciplines belonged to the field of environmental engineering; therefore, a large number of articles on circular agriculture are concentrated in natural science; that is, scholars have focused on research on technological progress, and there is little research involving the social sciences (such as economics). Before 2015, the total number of published papers in various disciplines did not show a large growth trend. After 2015, the research on circular agriculture in various disciplines decreased in some years, but it showed an explosive growth trend. The total number of published papers showed an exponential growth trend from 2017 to 2021, and the average growth rate reached 32.2%. The reason why research in this field is concentrated in natural science is that industrialization has brought disasters to the environment and resources, and more circular agricultural technologies must be developed to improve the utilization efficiency of resources.

### 4.3. Environmental Journals Pay More Attention to CA Research

We further analyzed the top five publications in terms of the number of articles published, aiming to explore which journals were mainly publishing articles related to circular agriculture and what preference topics were of those journals. The most published journal was *Sustainability*. On the one hand, this is because circular agriculture is an important model of sustainable development, which meets the requirements of the journal’s themes. On the other hand, it is because the total number of papers published in this journal was large. Among them, the two articles with the highest cited frequency were on the study of the transitional mode of the sustainable management of land resources [50] and the driving factors of circular agriculture [51]. The common point is that in addition to emphasizing the importance of circular technological progress, policies and market mechanisms play a prominent role in providing the direction of technological upgrading. The *Journal of Cleaner Production* had the highest impact factor among the five journals. Most articles focused on the study of circular economy from the perspective of product life cycle. The most cited article, up to 508 times, was on the recycling of food waste [52]. This article studied the recycling methods for various food wastes and put forward that the development of a recycling mode has serious management problems from economic and environmental aspects. The other three publications studied the role of biotechnology, chemical application and polymer science in circular agriculture, focusing on natural science.

### 4.4. CA Research Mainly Comes from China, the United States, Germany and Italy

We also analyzed the situation of countries, authors and institutions in the field of CA research in order to understand the development level, professional level of the authors and the key research directions of institutions in different countries. As Figure 3 shows, China had the largest number of publications, accounting for 42.3 percent of the total, and the top five authors and institutions were from China. An important reason for this is that China is the most populous country in the world, the contradiction between resource environment and population sustainable development is prominent, and there is an urgent need to develop recycling technologies to solve the problem of insufficient resources [53]. China focused on food science and technology, biochemistry and agricultural engineering, and such research continues to grow on a large scale. Figure 3 shows that the United States, India, Japan and Germany were closely linked to China. The main research in China and India focused on environmental science and energy science. In addition to environmental science, the United States, Japan and Germany gradually took economics and the social sciences into consideration [54].

Through the analysis of keywords, disciplines, publications and countries, it can be concluded that the current research on CA focuses on the natural science field dominated by technological upgrading. Although circular agricultural technologies emerge endlessly, there are problems of low conversion rates of scientific and technological achievements, and they are unable to meet the market demand, which not only generates sunk costs but also undermines the confidence of researchers. Therefore, it is necessary to develop appropriate science and technology according to market demand. It should be noted, however, that markets are not a fully self-managed area, which requires the use of designated policies to monitor and manage. The development of circular agriculture is a systematic problem, all things need to be tied together, which requires the country to adjust the market and social environment of CA from a long-term perspective.

## 5. Successes and Limitations of Circular Agriculture Projects under Different Political Backgrounds

After systematically reviewing literature, we found that current research on circular agriculture is concentrated in the natural sciences with a technological focus. However, the integration of market economics and policy into the research allows to obtain not only the dynamics of technological innovation in a circular agriculture economy but also the economic competitiveness. Therefore, we wanted to discuss studies from various social science disciplines. We tried to analyze the development of circular agriculture in different political and governance contexts by looking for evidence in the social field through practical cases.

### 5.1. The Cases of Circular Agriculture in Different Political and Governance Contexts

#### 5.1.1. Government-Led Agricultural Waste Utilization: The Case of China

The development of circular agriculture in China is typical of emerging economies and is a top-down development model. Karpenstein-Machan proposed a model called a “self-sustaining farm” [55], and examples of practical applications of this model in China include the “livestock–biogas–fruit” model [56] and the “pig–biogas–vegetable” model [57]. It should be noted that most Chinese circular agriculture farming is centered on biogas fermentation technology [58,59]. Therefore, in this section, we took China as an example. Anping County in Hebei Province and Shunyi District in Beijing were selected for our review. We found that under the guidance of the central government, many regions have helped to form a green recycling agriculture system through the resourcefulness of using manure and tailing vegetables.

Shunyi District mainly uses an “agricultural producers + operational service organizations + organic fertilizer processing plant + policy subsidies” operation mode. Through professional service organizations responsible for agricultural waste collection and transport, organic fertilizer processing plants are responsible for follow-up processing. Then, through the “government purchase services-villages and towns with the stacking-enterprise collection and hauling” operation mode, the government, through the form of operating subsidies, mobilizes the market to participate in the collection and hauling of agricultural waste enthusiasm so as to promote the long-term stable development of the comprehensive utilization of agricultural waste. The main practices can be summarized as follows: (1) Shunyi District, with the village as a unit, led by the government to set-up vegetable waste collection sites per 100 acres of vegetable park, was assigned to supervise and check the growers picking up the transport situation. Growers, in accordance with the requirements of the waste picking, stacked waste at the designated collection site. (2) Agricultural machinery professional cooperatives are responsible for collecting and transporting agricultural wastes, such as rotten leaves and fruits, to the organic fertilizer processing plant introduced by the government. (3) The organic fertilizer processing enterprise adopts the treatment process of “compound decomposing bacterium + turning over aerobic fermentation” to make organic fertilizer from vegetable plot waste in the whole area and to promote it.

Under the “Shunyi scheme”, organic fertilizer processing enterprises process 200 tons of organic fertilizer (30% water content) per day, which can consume 200 tons of vegetable waste (80% water content), 30 tons of dry straw (10% water content) and 200 tons of cow manure (65% water content). A processing site with an annual output of 40,000 tons can have an annual profit of CNY 7,024,000, which has good economic, social and ecological benefits.

In the initial stage of promotion, Shunyi District invested CNY 5 million in support funds. Since 2017, the district has invested approximately CNY 12 million annually to promote the scheme and subsidize all parties involved in the recycling of vegetable plot waste.

#### 5.1.2. Wastewater Recycling and Industry Association: The Case of Germany

Although Germany is a country “rich in water resources”, due to geographical climate change, water pollution and other reasons, there is indeed a shortage of water resources in some regions [60]. Therefore, agricultural wastewater reuse is receiving increasing attention as a strategy to support the transition to a circular economy for water and agriculture.

We selected the agricultural wastewater reuse scheme in Braunschweig (Germany) for our study. This reuse scheme combines the agricultural reuse of wastewater and sludge, crop production and bio-energy production [61]. The reuse program is managed by the Wastewater Association Braunschweig and aims at implementing large-scale agricultural wastewater irrigation systems in the region of Braunschweig. It is a complex network of linked activities with various environmental and economic benefits on a regional level [62]. The association’s members include the water board of the city of Braunschweig, the neighboring city of Gifhorn, and 90 farmers who own agricultural land in the association’s region.

In this system, the association owns the infrastructure for wastewater and irrigation, and the farmers own or lease the land for irrigated farmland. Farmers are autonomous entrepreneurs who decide on the crops grown and production methods based on market prices, yield expectations and personal business strategies. The interaction between the association and farmers regarding irrigated farmland crop cultivation is driven by the different interests of the farmers and the association. Farmers are interested in obtaining treated wastewater to supply their crops and to grow those crops that maximize profit. The association wants farmers to grow a variety of crops that will provide them with an adequate supply of wastewater. At the same time, the association is interested in providing farmers with sufficient wastewater. This is because from the farmers’ point of view, the stability of crop yields and the profitability of wastewater cultivation are necessary requirements for the continuation of the reuse program. The coordination of interests and the market drive result in good cooperation between farmers and enterprises. In other words, the provision of irrigation services is hierarchically coordinated with the spread of wastewater, and crop cultivation and crop selection are close to market governance. On-farm irrigation and wastewater irrigation connect the value chain of wastewater treatment and crop production, showing a hybrid governance structure [61].

#### 5.1.3. Circulation of Olive Farms: The Case of Italy

The olive tree is inextricably linked to the Mediterranean region, and olive cultivation is a typical agricultural activity in the Mediterranean basin. In the Mediterranean region, most olive groves are old plantations with low agrotechnical input intensity and often with limited expansion. Therefore, the economic benefits are marginal [63]. During olive processing, two main residual products are generally obtained: pomace and olive mill wastewater. After partial drying, both byproducts can be used as fuel to provide the energy (i.e., heat and power) needed in the milling process. In addition, the olive branches pruned during the maintenance of olive groves are an agricultural residue. These crop residues are usually collected in stockpiles, chopped and incorporated into the soil.

An olive farm in the Foggia District of the Apulia Region (Southern Italy) was selected as a case study with reference to existing the literature [63]. The farm’s olive groves cover approximately 10 hectares and are planted with old traditional olive varieties (such as “Coratina”, “Leccino” and “Peranzana”). The farm is able to grow olive trees and process olive oil independently and complete all steps of the agricultural cycle. The main farm practices can be summarized as: (1) Residue use: Olive forest (pruning) and olive oil extract residues are used as feedstock for pyrolyzing plants. The 70.0 tons of solid and semi-solid waste that may be generated during the 10 hectares of olive forest and grinding process can be converted into 13 tons of liquid fuel, 11 tons of biochar and 12 tons of gas fuel by pyrolysis, respectively [63]. (2) Bio-oil as fuel: The obtained pyrolysis oil (biooil) is used for power generation to meet the energy requirements of olive oil grinding operations and provide power for the dryer. The liquid and gaseous fuels produced in this rural area, in addition to meeting the energy needs for olive oil processing, also generate an electricity surplus, generating an additional income of approximately EUR 4,000 [63]. (3) Waste heat utilization: part of the waste heat of the dryer (steam) is recycled for drying pruning and refining. The remaining portion also provides the heat needed for olive oil processing. (4) Biochar as a soil amendment: Biochar obtained by pyrolysis is considered a relevant by-product and is used as an agricultural soil amendment to meet the needs of olive farms.

The challenges and opportunities go hand in hand in such a model. First, it is undeniable that the pyrolysis-biochar system contributes to the goal of a closed-loop and circular economy in agriculture as well as to the sustainability of the farm. At the same time, in terms of social benefits, it generates additional income for the farm operator and facilitates the operation. However, uncertainties remain regarding the adaptability of the biochar market. When farms are using this model, competition from other renewable energy sources (e.g., solar) needs to be considered, as well as competition with other waste management practices (e.g., pellet production).

### 5.2. Discussion and Analysis

The analysis of different countries helps us understand the practice of circular agriculture in a unique institutional setting (Table 1). However, we argue that some unique institutional model of CA cannot be widely replicated in different socioeconomic settings. For example, circular economy is promoted as a top-down national development strategy in China, but in many Western economies, it is a tool for redesigning bottom-up management policies [64]. In this section, we discuss the differences and commonalities between the ‘top-down’ and ‘bottom-up’ models. At the same time, from the perspective of social politics, we pointed out the differences in the role of government and its social impact in these cases.

#### 5.2.1. Top-Down Circular Agriculture Development Model

The Chinese practice confirms that the circular model is achievable in agriculture, and the government plays a key role in promoting CA [65,66,67]. These findings can inform public policy in emerging economies on similar development trajectories [68]. First, governments may cover some of the initial costs of CA projects. In this circular agriculture scheme, the government often supports the cost of these technology projects or introduces professional enterprises for biogas treatment. Second, the government can provide the professionals for circular agriculture. These professionals can help farmers with the necessary technical support and market information. On the one hand, farmers in China have relatively low education and lack experience in the specialized techniques of circular agriculture, making it almost impossible to meet the demand for adopting green technologies. On the other hand, the government can provide the service of organic products in order to help farmers obtain timely market information and reduce risks, thus, effectively adjust their production plans. Third, the government can fill the possible loopholes in the agricultural recycling system. For example, in partially circular agricultural systems, a portion of the organic fertilizer and the electricity produced by biogas fermentation can be delivered to grid companies and fertilizer markets for profit, which is an important way to obtain economic benefits from their circular operation. In this process, the government plays an important role in stabilizing the market for organic products and regulating their prices in order to motivate farmers to participate in circular agriculture.

For rural areas weak in economy, especially in developing countries, such as China, the initial investment in CA projects can be high, which requires policy and financial support such as investment subsidies [69] and targeted loans [70]. In other words, this top-down CA program emphasizes infrastructure development and financial subsidies. The model responds to both a strong central government-led food security policy and the concomitant goal of reducing carbon emissions from agriculture. In order to achieve both food security and agricultural carbon emission reduction goals, the CA project relies on government subsidies and incentives to promote the recycling of agricultural waste resources for local agricultural restructuring.

However, there are certain shortcomings and flaws in this model. First, this model may negate the goal of promoting demand reduction through carbon trading. Second, an elite-driven policy does not adequately address the different production goals of local institutions. This may be a common problem with top-down models of resource management programs and is consistent with findings from studies of water resources programs in China [71]. Third, this top-down implementation approach is costly and somewhat unstable in terms of human resources and management expenditures. The achievement of the expected results depends largely on the political interests of local bureaucracies shaped by central–local relationships. Thus, the duration and intensity of state intervention depends largely on the preferences of the current party leadership, rather than the legal system and market mechanisms. Its long-term viability depends on the willingness of the central government to provide financial transfers and to maintain their intensity. More complicated, CA project management is usually designed and implemented by national decision makers and external experts who occupy a hegemonic position. In other words, in the top-down mode, the decision-making power is still in the hands of technical elites. Their cooperation in circular agriculture strengthens existing organizational authorization and expands state power.

#### 5.2.2. Bottom-Up Circular Agriculture Development Model

We found that in areas with better agricultural infrastructure, the formation of a circular agriculture scheme led by enterprises or associations was also social in nature, while markets and laws play a key role in promoting CA. These findings can inform economies that are on similar development trajectories. For one, the starting point of the German and Italian recycling models was the cooperation between enterprises and farmers. It can be called the bottom-up scheme of circular agriculture. In this scheme, enterprises take on the basic aspects of technology research and development, facility construction, etc., while technology costs and infrastructure costs are often provided by enterprises after weighing the costs and benefits. The cost savings from sludge treatment and fertilization become a common interest for both the enterprise (association) and the farmer, which in turn leads to cooperation. Second, the bottom-up scheme needs to be supported by a more complete agricultural infrastructure and a better agricultural production scheme. The enterprise or association-led scheme of circular agriculture is basically found in countries with a high degree of agricultural development. On the one hand, farmers in these countries or regions have a higher willingness to participate in climate change response actions such as joining conservation associations and carrying out waste utilization activities. On the other hand, these countries or regions usually have a better level of agricultural infrastructure and technology such as the construction of pipelines for waste purification back to farmland or a higher level of smart agriculture development. The success of the bottom-up recycling agriculture model is inextricably linked to sound supporting laws and regulations, adequate financial support, flexible economic leverage and high-tech development. In the Italian and German schemes, theoretical research and practical innovation were more likely to arise due to the relatively higher level of farmers’ knowledge and were more orderly.

The CA programs under the business and association-led scheme emphasized a shift in consciousness and cooperation among enterprises and farmers. In this scheme, the choice is generally motivated by considerations and trade-offs of market interests. Companies or associations need to consider the cost of handling, transporting and hiring labor for agricultural waste, and farmers need to consider the cost of using the treated waste products versus the cost of buying them directly. This type of CA project is a re-innovation based on the agricultural production and rural employment already configured, rather than infrastructure construction as the project progresses.

However, there are certain shortcomings in this scheme. The enterprise-led circular agriculture development scheme also has certain drawbacks. Firstly, due to the difference of available resources, some farmers hold a pessimistic attitude towards circular agriculture. A study in southern Italy shows that half of farmers say they lack direct access to information about the circular economy and 40% complain about a lack of technical training [72]. Similarly, in a recent review of the same Dutch Environmental Assessment Agency [73], it was claimed that the ambitions for structural change in agriculture envisaged by the Dutch government have so far hardly been translated into concrete policy measures. The majority of farmers at the end of the adaptive range are critical of the Dutch Ministry of Agriculture’s CA ambitions, arguing that its circular farming program is inapplicable and unfeasible in practice [74]. Second, the sustainability of such a program will depend heavily on changes in the regulation of the agricultural waste utilization system. If agricultural waste use is legally restricted in the future, operators will have to seek alternative treatment technologies, which will require additional investments and increase the operating costs of agricultural waste treatment.

## 6. Social and Political Views on the Circular Agriculture System

Through the analysis of China’s, Germany’s and Italy’s circular agriculture, we found that complex and interrelated socio-political factors shaped the management of agricultural emission reductions (Figure 4).

Due to the different political and market developments, there are some differences in the models of developing circular agriculture in each country. In the case of Germany and Italy, their development relies on cooperation between farmers and enterprises, farmers and industry associations, with the government playing a regulatory role. This model emphasizes the importance of public participation and market regulation. However, the sustainability of the model depends heavily on changes in the institutional regulation of agricultural waste use. If agricultural waste use is legally restricted in the future and the cost–benefit of the recycling process changes, then the choices of farmers and firms may change. In other words, the emergence of this scheme is more likely to receive changes in economic efficiency. From the two cases in China, their circular development relies on the government, and local governments at all levels are the main implementer. This scheme emphasizes the importance of macroeconomic regulation. However, it is costly in terms of human resources and management expenditures, while suffering from the fact that its long-term viability depends on the willingness of the central government to provide financial transfers. In other words, this model is minimally influenced by market mechanisms and legal regulation and relies heavily on governing preferences. However, it is undeniable that under either model there are differences in their development patterns and directions due to the political, social and economic differences.

In agricultural development, the model adopted by a country is closely related to the national condition and the general international environment and is a product of economic development to a certain extent and the needs of a country’s citizens. After the modernization of agriculture, many developed countries began to pay attention to improving life quality and became alert to the shortage of resources and environmental pollution. They gradually adopted agricultural production methods that conserve resources and protect the environment and widely applied advanced industrial technologies to agricultural production, forming a model conducive to sustainable agricultural development in a relatively short time, i.e., the circular agricultural development model we advocate for now. In developing countries, represented by China, the circular economy is based on advanced experience in the process of industrialization when serious resource shortages and environmental disruption occur. In the process of agricultural modernization, when agricultural resources and environment appear similar to those of industry, people actively promote ecological agriculture while proposing the development of recycling-based agriculture. Therefore, there are some differences in the connotation of the circular system in China and the circular economy proposed by developed countries. Specifically, most developed countries are committed to waste disposal and environmental protection through a bottom-up model, initiated by markets and businesses. Developing countries, on the other hand, are committed to resource conservation and environmental protection while ensuring rapid economic development, facing the multiple pressures of resource shortages as well as waste disposal and environmental protection, and the development of the cycle relies on government guidance and support, which is a top-down model.

The complexity of circular agriculture management lies in its intersection with natural resources and human activities. As human activities increasingly affect the quantity of agricultural resources, risks such as carbon emissions and environmental pollution are further magnified, leading to more uncertainty regarding system outcomes and predictions. Traditionally, agricultural and environmental engineering perspectives treat circular agriculture as a technical and scientific problem (as analyzed in our literature review in Section 3) and tend to consider its solutions as dependent on the technical reliability of the operating system [75,76]. However, the main threats to nature caused by humans, such as climate change, require an approach that takes into account the interconnectedness of the social and natural world. Therefore, the perspective of the socio-political aspects of circular agriculture management should be integrated into the assessment of the sustainability of agricultural systems.

In fact, some developed agricultural countries have noticed the importance of socio-economic factors in circular agriculture. For example, studies on biogas in Germany have shown that in addition to techno-economic and biophysical factors, socioeconomic factors may also play a decisive role in the performance of future agricultural biogas plants [77]. The early literature also emphasized the role of government and institutions in the transition to sustainable development [78]. Particular emphasis has been placed on governments and institutions when examining the drivers of sustainability in emerging economies. These economies are often characterized by homogeneous institutions and constrained policy discourse, leading to government dominance in the transition to sustainability [79]. In general, democracy can create a favorable political and social climate for environmental activities and policies. Therefore, environment-oriented policies can be developed and implemented more effectively in this context [80]. In other words, the effectiveness of the overall environmental policy is crucial for the development of agriculture. This explains why some countries with a high degree of democracy have been able to successfully complete a system of circular agriculture, while others have not been able to or are still in the development stage.

Markets and economics influence the direction and sustainability of circular agriculture. Market imperfections are generally considered to lead to environmental pollution. However, environmental pollution is a source of important entrepreneurial opportunities [81]. As the world becomes more concerned about sustainability, the concept of “eco-business” has been introduced to advocate for policy and institutional changes. For “eco-entrepreneurs”, the challenge for organizations is to better integrate environmental performance into their economic business logic [82]. Even when examining some of the emerging bottom-up processes, researchers tend to focus on institutional actors such as governments, officials, regional governments, and research institutions [83]. For example, a two-year comparative case study of sustainable entrepreneurs in the Netherlands found that sustainable entrepreneurs created new symbols and theories, constructed new measures, built consensus and established new relationships to change or create new institutions, thus engaging in institutional change [84]. At the same time, sustainable entrepreneurs aim to introduce environmentally and socially friendly innovations to a large number of stakeholders [85]. To be clear, the purpose of focusing on these interest groups is not to promote the establishment of ecologically and economically viable enterprises but rather to advocate and promote changes in policy and institutional arrangements through the study of them.

In addition, the influence of interest groups in circular agriculture varies in different political contexts. For emerging economies, under state-set constraints, firms are the main driver of market-based transformation in emerging economies such as China; thus, entrepreneurship and business development can lead to sustainable transformation through fiscal incentives. At the same time, the imperfect institutional environment of emerging markets may paradoxically provide greater flexibility and opportunity for entrepreneurs to take action. A recent study suggests that entrepreneurship contributes more to environmental improvements in low-income countries than in high-income countries. Unfortunately, there is a general lack of attention to the role of entrepreneurship in promoting sustainable business in emerging markets [86].

## 7. Conclusions

Based on the analysis of the existing literature, we found that the research in the field of circular agriculture is concentrated in natural science, especially on technology development. Next, our examination of CA projects in China, Germany and Italy demonstrated the socio-political factors in the design and management of agricultural recycling systems under different political systems. We found that the top-down CA implementation approach emphasizes infrastructure development and relies heavily on government preferences and inputs. However, most publications on uncertainty in China’s agricultural system focused on the conflicting population densities and agricultural resources, underestimating the difficulty of managing costs and the sustainable inputs of the human costs in an authoritarian policy context. The management of circular agriculture in Italy and Germany is constituted by the different rationalities and interests of social actors at several levels. The bottom-up CA implementation approach values the response to market information, and its sustainability depends heavily on changes in the regulation of agricultural waste utilization systems. Compared to China, these publications on uncertainty in agricultural systems in developed countries take into account more socio-political factors in the educational, social and demographic fields. In summary, by highlighting the social relationships of farmers, businesses and national policies, socio-political perspectives show that circular technology and organizational control are constrained by institutional constraints in broader social structures and changing natural conditions in ecosystems. The progress of CA technology and infrastructure has caused ecological changes, affecting the availability of agricultural resources such as cultivated land, and further affecting the organization of implementing the distribution mechanism of technology and infrastructure. Therefore, our proposed socio-political perspective on circular management offers an alternative approach, as it focuses on the intersection of environmental and human systems.

This review shows that there are more complex and profound socioeconomic and political reasons for the development of circular agriculture worldwide. The regulation of politics, the development of society and the influence of the market have an impact on multiple subjects from the peasant level to the state level. With this review, we wanted to prove that policy frameworks based on any single discipline or single approach are largely ineffective. The most distinctive feature of the socio-political perspective, as opposed to natural science or other approaches, is its stronger subjectivity. As we found in our literature analysis, most of the existing research methods on circular agriculture are either experimental or empirical. However, in terms of case studies in different socio-political contexts, research in circular agriculture requires more attention to combine with the analysis of the research subjects. Thus, we believe that future research on circular agriculture should adopt more of a logical analysis and observational summary approach. For example, disciplinary, analytical and instrumental approaches can be used to explain the relationship between the development of circular agriculture and human activities. In the socially led scheme of sustainable agricultural management, a major challenge for companies and researchers is to develop business schemes and business environments that promote viable and acceptable production. Therefore, how to integrate human activities with the development of circular agriculture is a current challenge to be faced, and we need to assess it from a more comprehensive and integrated perspective. In summary, we urgently need to coordinate and make efforts in multiple socioeconomic and political fields together with the relevant actors.

Therefore, in future research, we strongly recommend an interdisciplinary collaboration that is not limited to the development of environmental technologies but further explores an integrated method to overcome the shortcomings of existing approaches.

## Figures and Tables

**Figure 1 ijerph-19-13117-f001:**
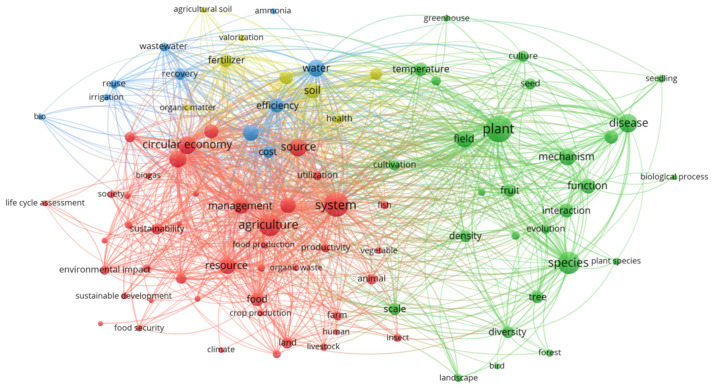
Keyword co-occurrence map of CA.

**Figure 2 ijerph-19-13117-f002:**
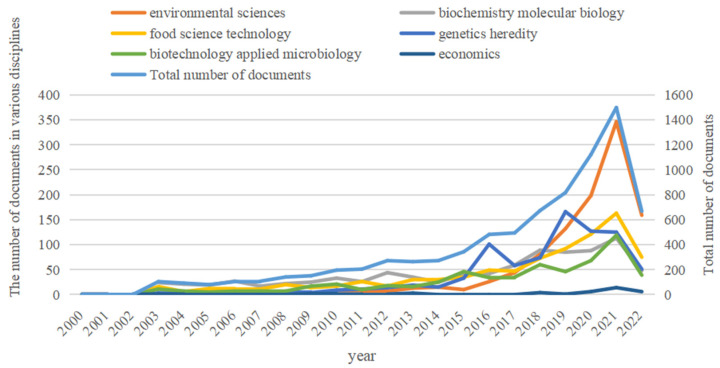
Discipline Distribution of CA research from 2000 to 2022 (June).

**Figure 3 ijerph-19-13117-f003:**
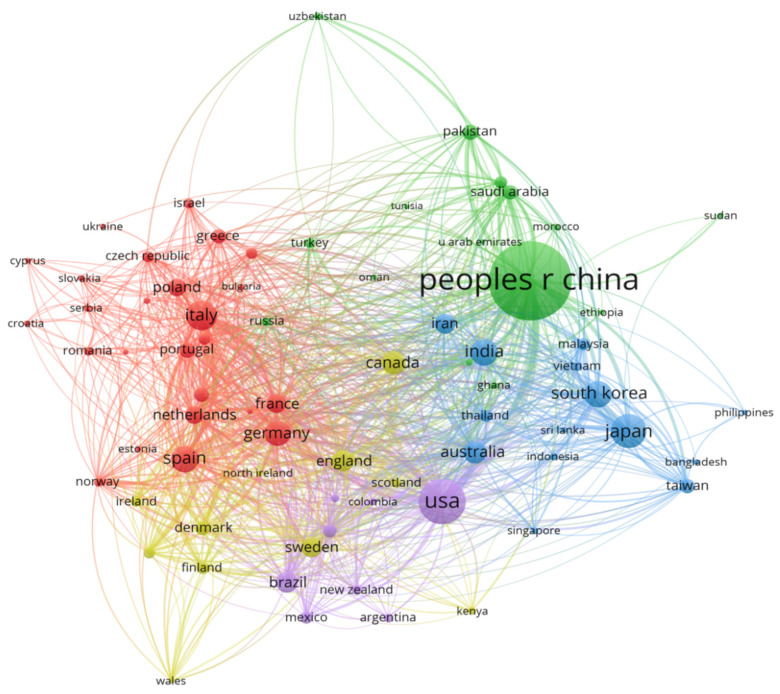
Countries co-occurrence map of CA.

**Figure 4 ijerph-19-13117-f004:**
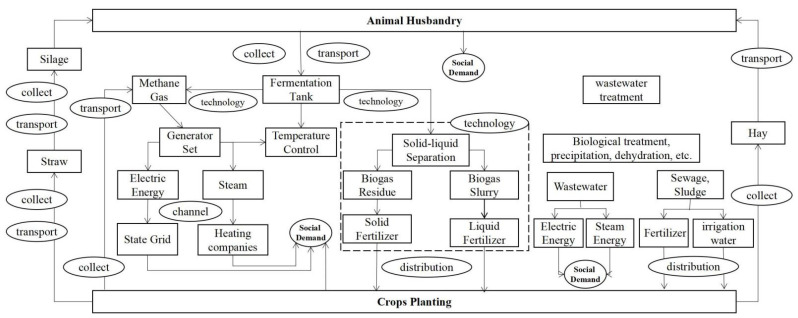
Circular agriculture development route diagram.

**Table 1 ijerph-19-13117-t001:** Cases of Circular Agriculture under Different Political and Social Backgrounds.

	Reuse of Agricultural Wastewater	Circulation of Olive Farms	Government-Led Waste Utilization
Case location	Braunschweig, Germany	Foggia district, Apulia region, Italy	Shunyi district, Beijing, China
Key role subject	Association	Farm	Government
Strength	Balancing the interests of the association and farmers	Contribute to the circular economy of the farm	Lower risk and comprehensive guarantee for farmers
Risk and challenge	Reliance on self-monitoring by industry associations Market fluctuations and legal constraints	Competition from other waste management practices Market fluctuations and legal constraints	Costly in terms of human resources and management expenses Policy making process driven by elite does not adequately address the different production goals of local institutions
Mode	Bottom-up	Bottom-up	Top-down

## Data Availability

Not applicable.

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
