# Peer review of "Toward a Socio-Political Approach to Promote the Development of Circular Agriculture: A Critical Review"

_ijerph, 2022, doi:10.3390/ijerph192013117_

Round 1

Reviewer 1 Report

In general, each section is too 'word-y' and needs to be shortened to keep the readers attention. The introductions suffers greatly from this as the it take far too long for the topic of the article to be introduced. There is plenty of relevant work discussed throughout the manuscript, however, the importance and impact of this is lost.

The quality of English language is not an issue here, however, a proof read is required as there are a small number of grammatical errors.

Reviewer 2 Report

Title: “Toward a socio-political approach to promote the development of circular agriculture: A Critical Review”

The title of the manuscript identifies the article's main point.

The actuality and relevance of the topic: the topic is relevant and timely.

Minor comments: Suggested Upgrade

The overall composition of the manuscript: The work is well-written and illustrated, with a good balance of relevant text and discussion. Shorten the introduction and keep the focus on the key points.

Line 117: the Background & methods chapter should be modified – separate them. The description of the background should go to the Introduction chapter, and the rest should be under the tile ‘Literature review’.

The framework and problems of the research are adequately explained. The graphs and tables are clear, accurate, and useful.

Use of research methodology; The strategies and methods used are appropriate.

Line 141: the authors mentioned ‘methods’ Why did you use the plural form? Are there more than one? The steps are not methods (e.g., case selection or identifying research questions are only the steps that you followed in your research project). What method(s) did you use? Secondary Data Analysis, comparative or mixed methods? Please clarify.

Lines 136-137: is the sentence clear? Which community? Did you mean scientific literature?

143: “these questions” Which questions do you mean? The following questions perhaps? The reference is not clear. Rephrase the paragraph.

Line 172: typo (in the field)

The acquired results logically support the conclusion.

Relevance of literature review; Overall, the review of previous literature is thorough, the latest and most up-to-date literature is mentioned.

 I hope the comments will help you finalise the paper. You could better highlight why the potential readers should read your paper, and why it is interesting for a wider audience.

Reviewer 3 Report

1.     Not read clearly and specifies the toward a socio-political approach to promote the development of circular agriculture, that the manuscript mentions.

2.     In the abstract it is mentioned that this review argues that circular agriculture projects are not only technical issues in the field of natural sciences, but also strongly influenced by social development. In the writing various objectives are written, in some cases they are not developed or are developed with no depth or analysis

3.     It is a very extensive writing on China and a few information dealing with two other countries (Italy and Germany).  The analysis should be homogeneous, as well as the discussion and results.

4.     It is very synthesized the section where show the important role of these factors through typical cases

5.     Tried to analyze the development of circular agriculture in different political and governance contexts by looking for evidence of the importance of social sciences in the social field through practical cases, but in this section it is only written about a single country.

6.     In the topic Bottom-up circular agriculture development model: the case of experience and Italy, I really couldn't read about it, only is written of China again without detailing what is said of Italy or Germany. In the methodology they chose these countries for analysis, however China is the central issue

7.     What is the difference between the discussion of point 4.1.2 and 4.2.3?

8.     As a target it is described that they tried attempt is made to propose a more critical solution from a socio-political perspective. It is not clear where it is written, neither something related to this is included in the conclusions.

9.     There is no good comparison between the countries analyzed

Round 2

Reviewer 3 Report

Considers that the authors adequately understood the comments